# Clinical Predictors of Mortality and Critical Illness in Patients with COVID-19 Pneumonia

**DOI:** 10.3390/metabo11100679

**Published:** 2021-10-02

**Authors:** Maamoun Basheer, Elias Saad, Rechnitzer Hagai, Nimer Assy

**Affiliations:** 1Internal Medicine Department, Galilee Medical Center, Nahariya 2210001, Israel; 2The Azrieli Faculty of Medicine, Bar-Ilan University, Safad 1311502, Israel; 3The Microbiology Lab, Galilee Medical Center, Nahariya 2210001, Israel; HagaiR@gmc.gov.il

**Keywords:** SARS-COV-2, NLR, high flow, BUN, insulin resistance, mortality, cytokine storm, predictors

## Abstract

Early identification of patients with COVID-19 who will develop severe or critical disease symptoms is important for delivering proper and early treatment. We analyzed demographic, clinical, immunological, hematological, biochemical and radiographic findings that may be of utility to clinicians in predicting COVID-19 severity and mortality. Electronic medical record data from patients diagnosed with COVID-19 from November 2020 to June 2021 in the COVID-19 Department in the Galilee Medical Center, Nahariya, Israel, were collected. Epidemiologic, clinical, laboratory and imaging variables were analyzed. Multivariate stepwise regression analyses and discriminant analyses were used to identify and validate powerful predictors. The main outcome measure was invasive ventilation, or death. The study population included 390 patients, with a mean age of 61 ± 18, and 51% were male. The non-survivors were mostly male, elderly and overweight and significantly suffered from hypertension, diabetes mellitus type 2, lung disease, hemodialysis and past use of aspirin. Four predictive factors were found that associated with increased disease severity and/or mortality: age, NLR, BUN, and use of high flow oxygen therapy (HFNC). The AUC or diagnostic accuracy was 87%, with a sensitivity of 97%, specificity of 60%, PPV of 87% and NPP of 91%. The cytokine levels of CXCL-10, GCSF, IL-2 and IL-6 were significantly reduced upon the discharge of severely ill COVID-19 patients. The predictive factors associated with increased mortality include age, NLR, BUN, and use of HFNC upon admission. Identifying those with higher risks of mortality could help in early interventions to reduce the risk of death.

## 1. Introduction

People infected with SARS-CoV-2 may develop COVID-19, which has a wide range of clinical severity, from a mild upper respiratory tract inflammation to a diffuse viral pneumonia causing acute respiratory failure, including lung injury, multiorgan failure, sepsis and death [1,2,3]. According to the NIH management guidelines, 80% of the COVID-19 patients worldwide were classified as mild (fever, cough, malaise), 14% severe (pneumonia and hypoxemia), and 5% as critical illness such as septic shock and acute respiratory distress syndrome (ARDS) [1,2,3].

Patients need support for many of the common complications of severe COVID-19, including pneumonia, hypoxemic respiratory failure, ARDS, sepsis, septic shock and acute kidney injury. In addition to the complications directly caused by COVID-19, there are complications from prolonged hospitalizations, which include secondary bacterial infections, thromboembolisms, gastrointestinal bleeding and critical illness [4,5,6,7,8]. Several therapeutic agents have been evaluated for treating COVID-19, such as the antiviral Remdesivir, steroids (e.g., dexamethasone and methyl-prednisone) and anti-IL-1 and anti-IL-6 inhibitors [9,10]. The average case fatality rate in China was 2.3%, but mortality was as high as 49% in patients with critical illness [11].

Many studies suggest a template or scores for predicting the prognosis of COVID-19 patients [12,13,14,15,16,17,18]. It is difficult to accurately predict clinical outcomes for patients across this spectrum of clinical presentations [12,13,14,15,16,17], thus making it difficult to manage these patients. An improved understanding of predictive factors for COVID-19 is critical for improving clinical decisions. If physicians can better identify those with higher mortality risks, interventions can be implemented to reduce the risk of death.

We analyzed clinical data of confirmed COVID-19 patients, to see whether we could find predictors of severity and mortality. A group of parameters was found to affect the severity and mortality of the patients. We also identified clinical features that contributed to the disease progression.

## 2. Results

### 2.1. Clinical Characteristics of the Surviving and Non-Surviving COVID-19 Patients

The data of 392 COVID-19 patients were analyzed. Of these patients, 318 were discharged and 74 died. The surviving group was significantly younger than the non-survivors (58 ± 18 vs. 78 ± 12, *p* < 0.001, respectively). Of these groups, 60 and 47% of the non-survivor and the surviving groups were males, respectively. Among the non-survivors, the BMI was significantly higher than in those that survived (Table 1). The majority of the non-survivors significantly suffered from hypertension, diabetes mellitus type 2, lung disease, hemodialysis and past medical use of aspirin. The majority of the non-survivors were diagnosed as having severe symptoms upon admission. No differences in the symptoms before admission (diarrhea, fever and dyspnea) were found between the two groups. The differences in the laboratory findings between the two groups included BUN, ALT, absolute lymphocyte count, neutrophil to lymphocyte ratio, triglycerides, inflammation markers (CRP, D-dimer and IL-6) and the percent of patients who suffered from ARDS, cytokine storm and insulin resistance, which was higher in the non-surviving group. Significant differences were seen in the use of O_2_ upon admission, the use of high flow in the hospitalization periods, mechanical ventilation and the need for O_2_ supply on day seven, which were higher in the non-surviving group. There were no differences in the COVID-19 treatments during the hospitalization time (Table 1).

### 2.2. Association between Risk Factors and Mortality

Risk factors such as age, gender, ethnicity, BMI and past medical history, including diabetes mellitus type 2, hypertension, lung disease, chronic kidney disease and hemodialysis, were studied for their correlation to mortality in the COVID-19 patients. The results show that being elderly and hemodialysis were the major risk factors for mortality in the COVID-19 patients (Table 2A). A multi-regression analysis of risk factors showed that age is the main risk factor. However, hemodialysis and a high BMI were also major risk factors for mortality in the COVID-19 patients (Table 2B).

Analysis of past medical history treatments showed that aspirin use is a major risk factor for COVID-19 mortality. Furthermore, the patients who received home oxygen supply before admission had a higher risk of mortality (Table 2C). No difference was found in the pre-admission symptoms such as fever, dyspnea and diarrhea, weakness, myalgia and saturation at admission.

### 2.3. Correlation between Blood Tests, Severity of Pneumonia and Mortality in COVID-19 Patients

The clinical parameters and blood tests upon admission of the COVID-19 patients included PAO2/FiO2, extent of lung injury (pneumonia), fibrinogen, D-dimer, platelet count, mean platelet volume (MPV), neutrophil to lymphocyte ratio (NLR), ferritin, ALT, AST, BUN, creatinine, sodium, phosphor, HDL and triglycerides. These parameters were tested for their correlation to mortality. Univariate analyses show a correlation between age, ARDS and NLR (Table 3A). Multi-regression analyses show significant correlations between age, NLR, BUN and high flow use upon admission and mortality in COVID-19 patients. The non-survivors were elderly, had high BUN, NLR and used high flow ventilation upon admission. These parameters were significantly associated with mortality in the severe COVID-19 patients (Table 3B).

### 2.4. Discriminant Analysis with Diagnostic Accuracy of the Correlations between Clinical Parameters and Survival

There is a good correlation between age, NLR, BUN and the use of high flow upon admission, and disease severity and mortality in the COVID-19 patients. The sensitivity, specificity, positive predictive value and negative predictive value are 97.9, 59.7, 86.7 and 91.4%, respectively (Table 4A,B).

### 2.5. Correlation between the 4-C Score, Cytokines Storm Syndrome, Liver Injury, ARDS and Mortality in COVID-19 Patients

Correlations between BUN classes, cytokine storm syndrome, PAo2/Fio2, 4-C score, liver injury and severity and mortality of the disease is shown in Figure 1. A high 4-C score shows good correlation with the cytokine storm syndrome and ARDS (Figure 1A) with the mortality being higher in this group. A high 4-C score and a decrease in the PAO2/FiO2 ratio correlates with high mortality (Figure 1B). Liver injury and high BUN upon admission correlates with the 4-C score and mortality, as seen in Figure 1C,D, respectively.

### 2.6. Kaplan Meir Survival Analysis According to HDL Classes

Kaplan–Meier analysis showed that the length of hospitalization is correlated with mortality in the COVID-19 patients. Long periods of hospitalization are a major risk of mortality (Figure 2). Survival analysis versus hospitalization length using the Cox regression confirms this relationship.

### 2.7. Pro-Inflammatory Cytokine Concentrations in COVID-19 Patients: Upon Admission vs. upon Discharge, Severe vs. Mild COVID-19 Patients, Survivors vs. Non-Survivors

Serum pro-inflammatory cytokine levels were measured upon admission and again upon discharge. CXCL-10, GCSF, IL-2 and IL-6 serum concentrations were significantly reduced upon discharge, as seen in Figure 3A. A comparison of these cytokines between the severely ill patients who died during their hospitalization period and the mildly ill patients who were discharged two days later was conducted. The concentrations of CXCL-10, IL-2 and IL-6 were significantly elevated in the non-surviving group (Figure 3B).

## 3. Discussion

Understanding the predictive factors for COVID-19 is critical for improving clinical decisions. Our study clearly indicates that age, NLR, BUN and use of high flow oxygen therapy upon admission are predictive factors associated with increased mortality in these patients. Identifying those with higher mortality risks can allow physicians to make early interventions and reduce the risk of death.

We found that age correlates with the severity of COVID-19 disease. Elderly patients are associated with elevated levels of hypertension and diabetes mellitus. This is due to stiffer vessels and impaired metabolic efficiency. This could explain the mechanism underlying the risk of being morbidly obese and elderly in COVID-19 [19,20,21,22]. These data are comparable with the data from the CDC about the severity of the disease in the elderly [19,20].

Our study also shows that high NLR correlates with the disease severity and mortality of the COVID-19 patients. The human immune response, induced by a viral infection, is primarily associated with lymphocytes. Systematic inflammation significantly reduces CD4+ T lymphocytes, and increases suppressive CD8+ T lymphocytes, thus significantly reducing cellular immunity [23]. For this reason, the inflammation caused by the virus increases the NLR [23]. A SARS-COV-2 virus infection may cause a massive reduction in the serum lymphocyte concentration, and therefore, a high NLR could be a predictor of disease severity.

High levels of blood urea nitrogen (BUN) upon admission showed a direct correlation with mortality in our COVID-19 patients. BUN is the end product of nitrogen metabolism and is partially reabsorbed from renal tubules [24,25,26,27]. Coronavirus can affect kidney function by entering kidney cells in a direct ACE2-dependent way and activate the renin-angiotensin-aldosterone system [26,27]. These systemic effects cause renal vasoconstriction, and consequently, glomerular filtration, causing a reduction in BUN excretion [26,27]. Additionally, high levels of inflammation in severe COVID-19 patients perturbs the renin-angiotensin system [27]. Therefore, BUN is a good marker of a high inflammation status and may be a good predictor of the severity of the disease.

A direct correlation was found between the use of high flow upon admission and a high mortality rate. The use of high flow means that the patient has a decreased lung capacity and an increased percent of lung injury with hypoxemia. It is also known that hypoxia, along with other high comorbidities, cause poor outcomes [28]. In the initial period of the disease, hypoxic respiratory failure was evaluated as typical ARDS [28]. However, other findings suggest that moderate-to-severely hypoxemic patients affected by COVID-19 may benefit from high flow use and could potentially decrease the need and duration of mechanical ventilation and ICU length of stay without a negative impact in hospital mortality [29]. This possibly may depend on the patients themselves and the intensity of the lung injury. For patients with broad pulmonary involvement and risk factors, as described above, HF is less likely to prevent ventilation, while those with low pulmonary involvement and without risk factors are more likely to improve.

The surviving group in our study were younger than the non-survivors and the majority of the non-survivors were overweight and male. The majority of the non-survivors significantly suffered from hypertension, diabetes mellitus type 2, lung disease, hemodialysis and were diagnosed as severe patients upon admission. Obesity is causally related to metabolic syndrome, which presents as hypertension, diabetes mellitus, coronary heart disease, stroke, renal disease and heart failure [21]. Obesity or excess ectopic fat deposition may be a unifying risk factor for severe COVID-19 infection, reducing protective cardiorespiratory reserve as well as causing immune dysregulation [20,21].

Inflammation markers (CRP, D-dimer and IL-6) were significantly higher in the non-survivor groups. The cytokine levels of CXCL-10, GCSF, IL-2 and IL-6 were also significantly higher upon admission of the severely ill patients and in the critically ill patients as well. The cytokine storm is the result of the immune response to the virus and it is a possible major mortality risk factor in COVID-19 patients. The mildly ill patients had low concentrations of these pro-inflammation cytokines. Accumulating evidence suggests that a subgroup of patients with severe COVID-19 develop this cytokine storm with high levels of pro-inflammatory cytokines and chemokines that are associated with pulmonary inflammation and extensive lung damage [30]. This increases the severity and the mortality of the disease [31].

The majority of the non-survivors in our study used aspirin before admission. The benefits of aspirin in critically ill COVID-19 patients are controversial. One study suggests that systemic anticoagulant usage reduces mortality in mechanically ventilated COVID-19 patients [32]. In ARDS, aspirin has been studied with mixed results, where some studies have demonstrated benefit and others have not [33,34,35,36,37,38]. Patients who have been using aspirin prior to being infected are patients who likely suffer from cardiovascular diseases, and therefore, a COVID-19 infection could be fatal due to the underlying cardiovascular disease and not the aspirin.

This study confirmed that the 4-C score is a good predictor for the severity of the disease. The 4-C score correlates well with the elevation in BUN, ARDS progression and liver injury. Elevated 4-C scores were correlated with mortality in the COVID-19 patients. The 4-C mortality score uses patient demographics, clinical observations, and blood parameters that are commonly available at the time of hospital admission and can accurately characterize the population of hospital patients at high risk of death [19].

Our study has some limitations. The data presented here were largely obtained from reports that emerged early during the COVID-19 pandemic. In addition, the wide diversity of study methodologies, statistical approaches, modest sample sizes and geographic sites may have confounded our interpretation of the data. Nevertheless, in the context of a severe pandemic caused by a novel virus, it is vital to address knowledge gaps in the field and identify factors that are potentially predictive of COVID-19 complications that warrant further investigation. Additional big studies of the COVID-19 risk predictors outside Israel should be completed.

## 4. Methods

### 4.1. Study Population

Electronic medical record (EMR) data from patients diagnosed with COVID-19 from November 2020 to June 2021, in the Galilee Medical Center COVID-19 Department, Nahariya, Israel, were used as the database. COVID-19 diagnosis was based on positive polymerase chain reaction (PCR)-based clinical laboratory testing for the SARS-CoV-2 virus. EMR data from 392 patients were analyzed. Of them, 318 were discharged and 74 died.

### 4.2. Study Design

A retrospective analysis of the EMR data from the COVID-19 patients was performed. The analyzed parameters were demographic background, past medical history and treatments, weight, BMI, symptoms before admission (fever, myalgia, dyspnea and diarrhea), blood laboratory tests (biochemistry, CBC, blood gases, blood type, coagulation tests and inflammatory markers), time to discharge or time to death, time to negative nasopharynges RT-PCR test and treatments upon hospitalization. A COVID-19 diagnosis was confirmed using positive real-time PCR assays from nasal and nasopharyngeal swab specimens. Cytokine storms were defined as described by Caricchio et al. [19]. Blood urea nitrogen (BUN) was classified into 3 groups, ≤20, 20–40 and ≥40 (mg/dL). The 4-C score was calculated as described by Knight et al. [18]. This score contains parameters such as age, sex at birth, number of comorbidities, respiratory rate on admission, peripheral saturation in room air, Glasgow coma scale, urea and CRP. The sequential organ failure assessment (SOFA) score was measured as individual scores for each organ, to determine progression of organ dysfunction.

### 4.3. Luminex-Based Multiplex Assay for Serum Cytokine Concentration

Serum cytokine concentrations were measured including CCL-2, CCL-3, CXCL-10, GCSF, IFN-gamma, IL-10, IL-2, IL-4, IL-6, IL-7 and TNFα. Blood samples of patients were collected upon admission and discharge. Altogether, an analysis of 15 samples was conducted. Samples from five severely ill patients upon admission and five samples of patients who died were compared to five samples of mildly ill patients.

Blood samples were withdrawn from mildly ill and severe patients upon admission and discharge for the mildly ill patients and incubated for 30 min at room temperature. After coagulation, the blood samples were centrifuged at 1500× *g* at 4 °C for 15 min and the serum was separated and aliquoted into 2-milliliter tubes and stored in a −80 °C freezer. For the cytokine testing, the serum was thawed on ice and serum was pipetted in cryotubes until the assay was performed. To assess serum cytokine levels, the human high sensitivity cytokine Luminex custom 8-plex kits (R&D Systems, Inc., Minneapolis, MN, USA) were used. Testing was performed in 96-well plates according to the manufacturer’s instructions. Test samples were run in singles, while standard samples were run in duplicates. In brief, color-coded superparamagnetic beads coated with analyte-specific antibodies were utilized using the Luminex assay. Beads that recognize different target analytes were mixed together and incubated with the serum sample. Captured analytes were subsequently detected using a cocktail of biotinylated detection antibodies conjugated to streptavidin-phycoerythrin. The magnetic beads were then isolated and measured using the Luminex MAGPIX^®^ Analyzer (R&D Systems, Inc., Minneapolis, MN, USA).

### 4.4. Ethics

This study was approved by our medical center’s local ethics committee (N 231-20). Retrospective analysis of data from our electronic medical record database was performed under the oversight of the ICH guidelines for good clinical practice.

### 4.5. Outcomes

We defined severity on admission according to the 4-C score. A 4-C score above 10 was considered severe and accounted for a 30% probability of death. We defined critical COVID-19 illness as a composite of admission to the intensive care unit, invasive mechanical ventilation, ARDS or death.

### 4.6. Statistical Analysis

Statistical analysis was performed using the WinSTAT program. Results are presented as mean + SD for continuous variables. For categorical variables, the frequency and corresponding diagnosis percentage are provided. The Spearman test was used for correlations between two quantitative variables. Univariate direct regression analysis and multivariate stepwise regression analysis were performed for individual variables, including clinical and biochemical variables as an independent variable, and survival or death as the dependent variable. The diagnostic validity of the powerful predictors (age, BUN, neutrophil to lymphocyte ratio (NLR), 4-C score, high-flow) was tested regarding correct classification of death using discriminant analysis. The percentage of cases correctly classified by each diagnostic test as well as the performance measures of sensitivity, specificity, and positive and negative predictive values were calculated using discriminant analysis. Survival analysis versus hospitalization length according to Cox regression, and Kaplan–Meier survival analysis according to HDL classes, were recorded. Tests of significance were two-tailed, with a significance level of less than 0.05. WinSTAT is the statistical add-on program for Microsoft Excel (Kalmia Co., California, MA, USA).

## 5. Conclusions

Many clinical characteristics are associated with increased disease severity and/or mortality, including age, NLR, BUN, and use of high flow oxygen therapy. Identifying those with a higher risk of mortality could help in making the right interventions during admission and reduce unwanted outcomes.

## Figures and Tables

**Figure 1 metabolites-11-00679-f001:**
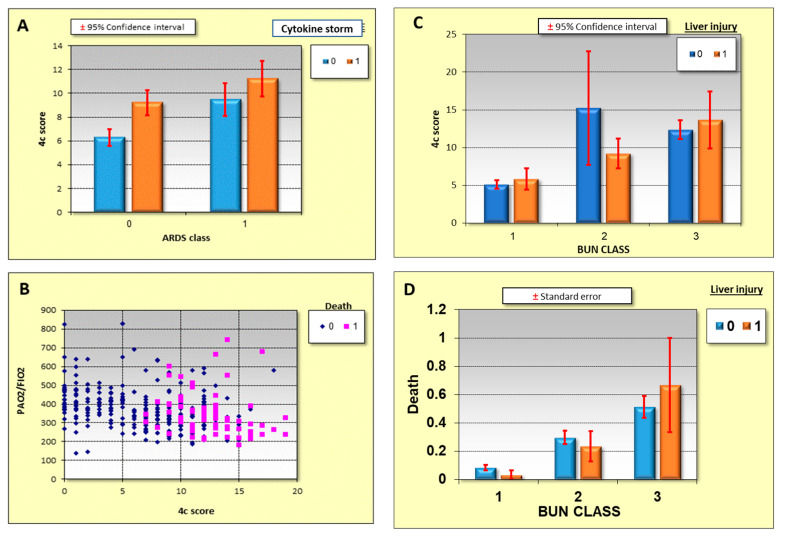
Correlations between ARDS, liver injury, PAO2/FiO2, the 4-C score, and mortality in COVID-19 patients. (**A**): Correlation between the 4-C score, cytokine storm syndrome and ARDS. (**B**): Correlation between the 4-C score, liver injury and BUN classes. (**C**): Correlation between the 4-C score and a decreased PAO2/FiO2 ratio and high mortality. (**D**): Correlation between liver injury, BUN class and high mortality. Standart deviation added to (**A**,**C**,**D**).

**Figure 2 metabolites-11-00679-f002:**
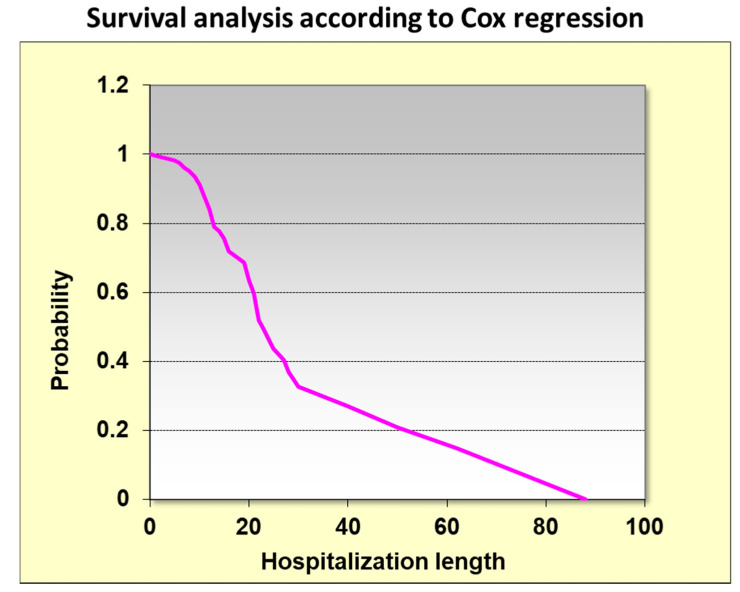
Survival analysis versus hospitalization length using Cox regression.

**Figure 3 metabolites-11-00679-f003:**
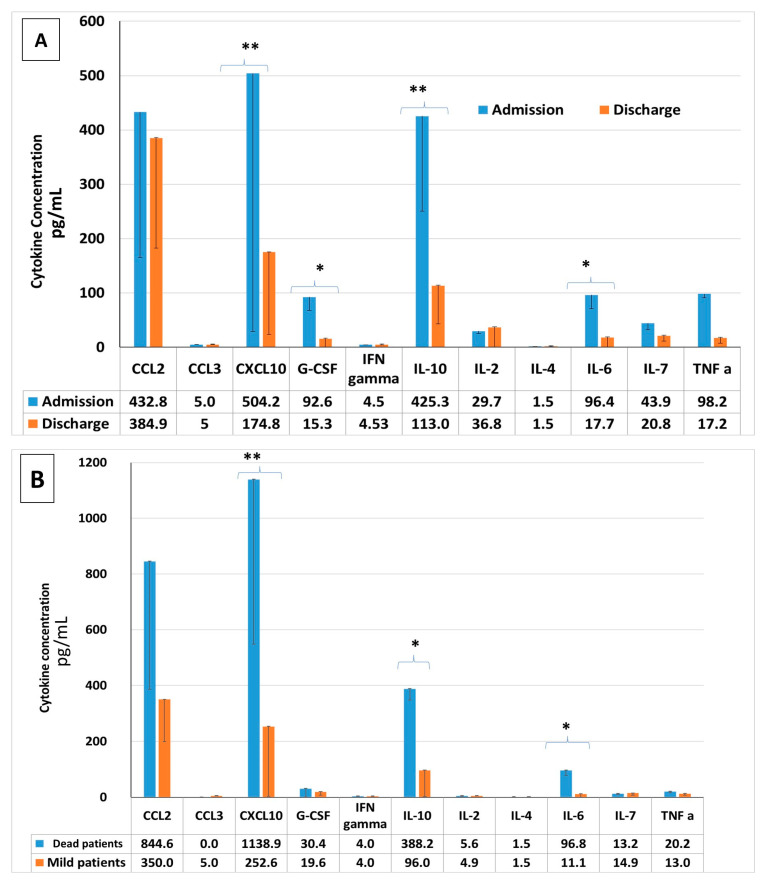
Cytokines levels in COVID-19 patients: The cytokines measured are CCL-2, CCL-3, CXCL-10, GCSF, IFN-gamma, IL-10, IL-2, IL-4, IL-6, IL-7 and TNFα. Cytokine levels were measured upon admission and upon discharge, in severely ill and mildly ill patients, and also between survivors and those not surviving. (**A**): Cytokine levels upon admission vs. upon discharge. (**B**) survivors vs. non-survivors. * *p* < 0.05, ** *p* < 0.001.

**Table 1 metabolites-11-00679-t001:** Clinical characteristics of the surviving and non-surviving patients with COVID-19 infection. NS: not significant (*p* value < 0.05).

Variable	Survivors	Non-Survivors	*p*-Value
Total	*N* = 318	*N* = 74	
Age	58 ± 18	78 ± 12	0.001
Male (%)	47%	60%	600
BMI	29 ± 6	32 ± 11	0.008
**Comorbidities %**			
Diabetes (%)	30	57	0.001
Hypertension (%)	48	84	0.001
Lung disease (%)	9	18	0.03
Hemodialysis (%)	5	15	0.003
Aspirin use (%)	30	61	0.001
**Symptom’s duration before admission to hospitals (days)**	6 ± 5	9 ± 12	0.19
**Symptoms before admission** **(% of total)**			
Fever %	57	56	0.77
Diarrhea %	6	2	0.26
Dyspnea %	59	59	0.74
Clinical severity on admission %	30	54	0.001
**Lab Findings upon admission**			
Hemoglobin (mg/dL)	13 ± 3	12 ± 1.2	0.06
Absolute neutrophil count (×10^3^/µL)	13 ± 9	13 ± 6	0.9
Absolute lymphocyte count (×10^3^/µL)	2.06 ± 11	1.4 ± 4	0.001
Neutrophil to lymphocyte ratio (NLR)	7.01 ± 3	9.1 ± 0.8	0.001
Platelet (×10^3^/µL)	220 ± 86	220 ± 90	0.79
BUN (mg/dL)	19 ± 14	40 ± 27	0.001
Creatinine (mg/dL)	2 ± 8	2.2 ± 2	0.7
Triglycerides (mg/dL)	145 ± 148	157 ± 48	0.08
HDL (mg/dL)	33 ± 12	30 ± 13	0.09
Insulin Resistance (TG/HDL)	5.4 ± 5.8	6.1 ± 5.0	0.02
C-reactive protein (CRP) (mg/dL)	68 ± 78	109 ± 97	0.001
Ferritin	632 ± 987	1041 ± 3594	0.57
D-dimer	1664 ± 3513	2539 ± 3374	0.001
Fibrinogen	641 ± 183	660 ± 180	0.33
IL-6	34 ± 52	264 ± 111	0.001
ALT	23 ± 21	37 ± 51	0.001
Cytokine storm (% of total)	25	51	0.001
4-C score	8 ± 20	12 ± 3	0.001
SOFA score	1.3 ± 1.4	2.7 ± 2.2	0.001
O_2_ supplement on admission %	55	100	0.001
High flow use (% of total)	14	82	0.001
Mechanical ventilation (% of total)	2	46	0.001
ARDS (% of total)	17	34	0.006
O_2_ supply at day 7 (% of total)	18	99	0.001
Hospitalization length	9.5 ± 10	17.2 ± 13	0.001
Time to negative PCR	17 ± 8	20 ± 8	0.16
**Treatment in hospitalization**			
***Steroid therapy* (% of total)** Methylprednisolone Dexamethasone	93 86 14	97 90 10	0.26 0.75 0.75
LMWH (% of total)	96	99	0.34
Remdesivir (% of total)	12	19	0.21
Vitamin D (% of total)	97	97	0.9

**Table 2 metabolites-11-00679-t002:** Correlations between risk factors and mortality in COVID-19 patients. (**A**): Univariate analysis of the strength of risk factors in mortality prediction. (**B**): Multiregression analysis of risk factors. (**C**): Analysis of past medical history of the patients and mortality risks.

**A**					
	**Coefficient**	**95%Conf. (±)**	**Std. Error**	**T**	***p*-Value**
** *Constant* **					
**Age**	0.01	0.002	0.001	5.4	0.00001
**Gender**	−0.04	0.07	0.036	−1.36	0.17
**Ethnicity**	−0.04	0.07	0.03	−1.23	0.21
**BMI**	0.001	0.005	0.002	2.23	0.02
**DM**	0.04	0.088	0.044	1.05	0.2
**HTN**	−0.02	0.048	0.048	0.5	0.6
**Hemodialysis**	0.1	0.07	0.07	2.2	0.02
**B**					
	**Coefficient**	**95%Conf. (±)**	**Std. Error**	**T**	***p*-Value**
** *Constant* **					
**Age**	0.01	0.001	0.001	7.7	0.00001
**BMI**	0.01	0.005	0.0025	2.3	0.00001
**Hemodialysis**	0.176	0.13	0.069	2.5	0.00001
**C**					
	**Coefficient**	**95%Conf. (±)**	**Std. Error**	**T**	***p*-Value**
** *Constant* **					
**Past Aspirin use**	0.14	0.08	0.04	3.6	0.0003
**O_2_ supplement before admission**	0.25	0.08	0.04	6.2	0.00001

Abbreviations: BMI body mass index; DM, type 2 diabetes mellitus; HTN, hypertension. Abbreviation: SE are the standard errors of the regression coefficients. T is the quotient of the coefficient. Two-sided *p* values or observed significance levels.

**Table 3 metabolites-11-00679-t003:** (**A**): Univariate analyses of the strength of clinical and laboratory tests in predicting mortality in COVID-19 patients. (**B**): Multiregression analyses of the clinical parameters, blood tests and their correlation to mortality in COVID-19 patients.

**A**					
	**Coefficient**	**95%Conf. (±)**	**Std. Error**	**T**	***p*-Value**
** *Constant* **					
**Age**	0.01	0.01	0.008	1.35	0.18
**BMI**	0.008	0.024	0.01	0.89	0.38
**SO_2_ upon admission**	−0.004	0.044	0.02	−0.26	0.79
**% Pneumonia**	0.0003	0.007	0.003	0.104	0.9
**PAO_2_/FIO_2_**	0.0002	0.001	0.0004	0.42	0.67
**ARDS class**	0.187	0.37	0.18	1.043	0.3
**D-Dimer**	−0.0001	8.6	0.001	−0.50	0.62
**Fibrinogen**	−0.000	0.0007	0.001	−1.26	0.21
**NLR**	0.00125	0.0014	0.001	1.84	0.076
**Ferritin**	0.001	0.0001	0.001	0.75	0.45
**IL-6**	0.0007	0.002	0.001	0.54	0.58
**ALT**	−0.001	0.007	0.001	−0.51	0.61
**BUN**	0.004	0.014	0.007	0.56	0.56
**BUN Class**	0.096	0.36	0.175	0.557	0.58
**Cytokine storme**	−0.068	0.33	0.16	−0.427	0.67
**CRP**	0.0001	0.002	0.001	0.1	0.92
**HDL**	−0.006	0.016	0.008	−0.81	0.42
**HDL Class**	0.003	0.34	0.169	0.01	0.98
**TG**	−0.0004	0.002	0.0014	−0.29	0.76
**TG/HDL**	−0.0004	0.055	0.026	−0.017	0.98
**4C-score**	−0.012	0.07	0.03	−0.36	0.71
**High flow use**	0.18	0.38	0.18	0.96	0.34
**B**					
	**Coefficient**	**95%Conf. (±)**	**Std. Error**	**T**	***p*-Value**
** *Constant* **					
**Age**	0.009	0.005	0.002	3.1	0.003
**NLR**	0.001	0.001	0.0005	2.3	0.022
**BUN**	0.006	0.004	0.002	2.9	0.004
**High flow use**	0.258	0.183	0.091	2.8	0.006

Abbreviation: SE are the standard errors of the regression coefficients. T is the quotient of the coefficient. Two-sided *p* values or observed significance levels.

**Table 4 metabolites-11-00679-t004:** The validity (predictive power) of age, NLR, BUN, and use of HFNC, and mortality in COVID-19 patients. The accuracy of the calculations is 87%. (**A**): The number of samples: Predicted condition—218, survival; 47, death; and true condition, 0 for disease and 1 for no disease. (**B**): the sensitivity, specificity, positive predictive value and negative predictive value.

**A**		
**Actual count**	0	1
218	189	29
47	4	43
**B**		
Specificity		59.7
Sensitivity		97.9
Positive predictive value		86.7
Negative predictive value		91.5

## Data Availability

The data presented in this study are available on request from the corresponding author. The data are not publicly available due to privacy of the patient.

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
