# Peer review of "Clinical Predictors of Mortality and Critical Illness in Patients with COVID-19 Pneumonia"

_metabolites, 2021, doi:10.3390/metabo11100679_

Round 1
Reviewer 1 Report
The authors investigated the clinical predictors of mortality in COVID-19 patients. Data from electronic medical records (EMR) of patients undergoing COVID-19 treatment at the Galilee Medical Center COVID-19 Department (Israel) was used as a database. The diagnosis of COVID-19 was based on positive clinical studies based on the polymerase chain reaction (PCR) for the SARS-CoV-2 virus.
The study population was quite small and ultimately amounted to 240 people. Nevertheless, the authors indicated prognostic factors associated with increased mortality in COVID-19 patients, such as: age, NLR, BUN and the use of HFNC.
The article brings many new observations on the course of the COVID-19 disease and can be a valuable source of relevant information in the fight against the SARS-CoV-2 virus.
Author Response
- EMR data from 392 patients were analyzed. Of them, 318 were discharged and 74 died (line 254).
Reviewer 2 Report
Dear authors and editors, thank you for asking me to review this. I think this is a well written article and obviously a lot of thought has gone into this - this describes a cohort of patients, and the results are not that different from what is already known. The methods and results are extensively described and correct
What do the authors think is the novel element of their study? That is the major selling point of their article, which I think is missing.
Only one specific comment is that the term Arab/Jews could be misconstrued as a racist connnotation. I would strongly suggest that this is changed to another term.
The methods section should also appear earlier on in the manuscript and not at the end
Overall, I would recommend for publication if the above points can be sorted
Author Response
-We think that the new predictors like BUN and NLR are novel elements that could predict the severity of the patients on admission. The cytokine levels of CXCL-10, GCSF, IL-2 and IL-6 also could be a good predictor of the severity of the patient on admission.
-The term Arab/Jews was changed
-The methods section arranged as the journal instruction order.